# Clove Oil-Nanostructured Lipid Carriers: A Platform of Herbal Anesthetics in Whiteleg Shrimp (*Penaeus vannamei*)

**DOI:** 10.3390/foods11203162

**Published:** 2022-10-11

**Authors:** Somrudee Kaewmalun, Teerapong Yata, Sirikorn Kitiyodom, Jakarwan Yostawonkul, Katawut Namdee, Manoj Tukaram Kamble, Nopadon Pirarat

**Affiliations:** 1International Graduate Course of Veterinary Science and Technology (VST), Faculty of Veterinary Science, Chulalongkorn University, Bangkok 10330, Thailand; 2Unit of Biochemistry, Department of Physiology, Faculty of Veterinary Science, Chulalongkorn University, Bangkok 10330, Thailand; 3Wildlife, Exotic and Aquatic Animal Pathology Research Unit, Department of Pathology, Faculty of Veterinary Science, Chulalongkorn University, Bangkok 10330, Thailand; 4National Nanotechnology Center (NANOTEC), National Science and Technology Development Agency (NSTDA), Pathumthani 12120, Thailand; 5Department of Anatomy, Faculty of Science, Mahidol University, Bangkok 10400, Thailand

**Keywords:** *Penaeus vannamei*, anesthesia, biodistribution, clove oil, nanostructured lipid carriers, toxicity

## Abstract

Whiteleg shrimp (*Penaeus vannamei*) have been vulnerable to the stress induced by different aquaculture operations such as capture, handling, and transportation. In this study, we developed a novel clove oil-nanostructured lipid carrier (CO-NLC) to enhance the water-soluble capability and improve its anesthetic potential in whiteleg shrimp. The physicochemical characteristics, stability, and drug release capacity were assessed in vitro. The anesthetic effect and biodistribution were fully investigated in the shrimp body as well as the acute multiple-dose toxicity study. The average particle size, polydispersity index, and zeta potential value of the CO-NLCs were 175 nm, 0.12, and −48.37 mV, respectively, with a spherical shape that was stable for up to 3 months of storage. The average encapsulation efficiency of the CO-NLCs was 88.55%. In addition, the CO-NLCs were able to release 20% of eugenol after 2 h, which was lower than the standard (STD)-CO. The CO-NLC at 50 ppm observed the lowest anesthesia (2.2 min), the fastest recovery time (3.3 min), and the most rapid clearance (30 min) in shrimp body biodistribution. The results suggest that the CO-NLC could be a potent alternative nanodelivery platform for increasing the anesthetic activity of clove oil in whiteleg shrimp (*P. vannamei*).

## 1. Introduction

The whiteleg shrimp *(Penaeus vannamei)* has become one of the most economically valuable aquaculture species in the world because of their fast-growing nature and high tolerance level to density, salinity, and temperature change. In 2020, the global production of whiteleg shrimp was 5.81 million tons [1]. However, low stress tolerance, poor survival, and low disease resistance ability have impacted sustainable whiteleg shrimp production [2]. Unfortunately, stress-induced simulators such as handling techniques, diverse environmental variations, and live transportation have been incorporated throughout the cultivation process [3]. Stress has negative impacts on animals, resulting in massive morbidity and mortality due to impaired immune response and increased disease susceptibility [4,5]. Furthermore, shrimps territorial and cannibalistic behavior contributed to low survival rates during transportation [6], which could result in significant economic loss for the shrimp farming industry [7].

Most operations in shrimp farming are conducted without anesthetics, but live transportation of shrimp can be challenging due to their quick movement, cannibalistic behavior, and sharp rostrum [8]. Furthermore, numerous live transportation technologies have been developed to improve animal well-being standards and the survival rates during delivery from farms to markets. Importantly, various drugs, including benzocaine, quinaldine, 2-phenoxyethanol (2-PE), and tricaine methanesulfonate (MS-222), have been frequently employed for anesthesia in aquaculture [9]. Herbal anesthetics such as basil, thyme, mint, rosemary, lavender, citronella, verbena, and camphor have been used in aquaculture, although clove oil is the most often employed [10]. In Asia, essential oils from *Mentha piperita* [11], *M. spicata* [12], *Eucalyptus* sp., and *Origanum* sp. [13], and *Syzygium aromaticum* [14,15] have been found to induce anesthesia in fish and shrimp with positive health effects. *M. piperita* (Europe and the Middle East), *M. spicata* (Europe and Asia), *Origanum* sp. (south-west Asia), *S. aromaticum* (south-east Asia), and *Eucalyptus* sp. (Australia) are available in their native areas and have been adapted to many regions of the world [16]. 

The essential oil extracted from cloves *(Syzygium aromaticum*) is preferable to synthetics since it is more environmentally friendly and cost-effective [17]. The active ingredients of clove oil (CO) are eugenol (4-allyl-2methoxyphenol) and isoeugenol (4-propenyl-2-methoxyphenol), which have fewer negative effects on animals as well as less harmful substances retained in the meat [18]. Previous studies have demonstrated that CO is effective and safe for *P. vannamei* [19], *Macrobrachium rosenbergii* [20], *Palaemonetes sinensis* [21] and *P. monodon* [22].

Chemical anesthetics have limited application owing to safety issues in human beings and fish [23]. In contrast to certain anesthetics such as MS-222, clove oil anesthesia does not need a withdrawal phase since it is rapidly excreted from blood and tissues [24]. Consequently, clove oil seems to be a better alternative in commercial aquaculture operations, where anesthetics may be employed in substantial quantities by unskilled workers and discharged into natural water bodies. Clove oil, on the other hand, is insoluble in water. Prior to use for emulsification, it should be mixed with ethanol, which could be harmful to fish and other aquatic animals. Therefore, we have successfully developed and evaluated a safe, cost-effective, and easy-to-use form of clove essential oil by converting the poorly water-soluble CO into a soluble nanoparticulate platform for tilapia anesthesia [25]. There are three major types of lipid nanoparticles: nanoemulsions, solid-lipid nanoparticles (SLNs), and nanostructured lipid carriers (NLCs). Nanoemulsions are the former structures of lipid-based nanoparticles. Recently, the SLNs and NLCs have been modified to improve the nanoemulsion delivery system. Although various carriers have been formulated and studied, the NLCs have been recognized as superior lipid-based carrier systems to others [26]. We proposed a new generation of clove oil in the form of NLCs that would be suitable as an anesthetic in shrimp.

In this study, we aimed to determine the physicochemical characterization of NLCs followed by in vitro drug release. Furthermore, the appropriate concentration of CO-NLCs for anesthetic induction and recovery time, as well as the toxicity and biodistribution of CO-NLCs in shrimp, was optimized.

## 2. Materials and Methods

### 2.1. Source of Essential Clove Oil and Chemicals 

The animals and protocol of this study were officially approved by the Institutional Animal Care and Use Committee of the Faculty of Veterinary Science, Chulalongkorn University (IACUC: 2031018). All procedures were carried out according to the university’s guidelines and regulations, as well as policies governing biosafety procedures. Pure essential clove oil was purchased from Thai-China Flavors and Fragrances Industry Co., Ltd. (Phra Nakhon Si Ayutthaya, Thailand). Polysorbate 20, sorbitan oleate 80, and glycerol were from Croda, Thailand. Cetearyl alcohol and coco-glucoside were supplied by Chemico Inter Corporation, Thailand.

### 2.2. Schematic Overview of the Experimental Program

Prior to use as an anesthetic, clove oil should be mixed with ethanol, which may be detrimental to aquatic animals after discharging into water bodies. Moreover, it could be a skin allergy to unskilled labor. Therefore, we proposed a safe, cost-effective, and easy-to-use form of clove essential oil by converting the poorly water-soluble CO into a soluble novel nanodelivery platform of herbal anesthetics in whiteleg shrimp (Figure 1). To achieve this, we have formulated clove oil-nanostructured lipid carriers (CO-NLCs), followed by characterization, in vitro drug release capability, and stability tests. Importantly, the appropriate dose for anesthetic effect and biodistribution were investigated in the shrimp body as well as the acute multiple-dose toxicity study. CO-NLCs could be a good alternative nanodelivery platform for increasing the anesthetic activity of clove oil in whiteleg shrimp.

### 2.3. Preparation of CO-NLCs

The formulation of CO-NLCs was prepared by the high-speed homogenization method [27] with slight modifications. Briefly, the clove oil (20 g) was incubated in a water bath at 70 °C. As an oil phase, 3 g of Sorbitan Oleate 80 and 2 g of cetearyl alcohol and coco-glucoside were added to warmed clove oil, respectively. A mixture of polysorbate 20 (3 g), Glycerol (2 g), and DI water (70 g) was incubated at 70 °C until an aqueous phase formed. A pre-emulsion was mixed by adding the warmed aqueous phase into the oil phase and stirred at 300 rpm for 10 min. Lastly, clove oil was emulsified by a high-speed homogenizer at 10,000 rpm for 10 min and kept at room temperature until further use.

### 2.4. Physicochemical Characterization of Clove Oil NLCs

The formulation of CO-NLCs was evaluated for particle size, polydispersity index (PDI), and zeta potential by dynamic light scattering (DLS) (Nanosizer, Malvern, UK). The measurements were performed in triplicate with particle suspensions diluted in distilled water (a ratio of 1:100) at room temperature. The morphology of CO-NLCs was examined by transmission electron microscopy (TEM) (JEM-2100 plus, JEOL, Japan). Before the analysis, the CO-NLCs were dispersed in distilled water. A drop of nanoparticle suspension was placed on a carbon-coated copper grid followed by drying. Images were taken at 200000× magnification. 

### 2.5. Physical Stability Study

The accelerated stability study was carried out to determine the shelf life of the CO-NLCs [28]. The samples were kept in tightly sealed glass bottles and stored at two different temperatures and ambient humidity conditions (30 °C/60% and 40 °C/75%) for a period of 3 months. The physical stability was analyzed using DLS at 0, 1, 2, and 3 months to determine the particle size and zeta potential.

### 2.6. In Vitro Drug Release Study

The HPLC technique was used to determine the encapsulation efficiency (EE) and release profile of eugenol [25]. The drug release study was performed to evaluate eugenol release from the STD-CO and CO-NLCs. The amount of drug trapped in the formulation can be calculated and expressed as the EE. All measurements were performed in triplicate. The EE can be measured by using the following equation:%EE = [(C_i_ – C_f_)/C_i_)] × 100(1)

C_i_ represented the initial concentration of eugenol in the CO-NLCs; C_f_ represented the concentration of unencapsulated eugenol.

### 2.7. Culture Environment

The whiteleg shrimp (*P. vannamei*) with an average weight of 3 g were purchased from a commercial shrimp farm in Samut Songkhram, Thailand and transferred to the aquaculture laboratory. Shrimps were distributed equally into 30 tanks (300-L capacity) and fed twice a day with commercial feed (INTEQC Feed Co., Ltd., Samut Sakhon, Thailand) at the rate of 2% body weight. During the trial, water quality parameters such as temperature (28 °C), pH (8.2), DO (6 mg/mL), and salinity (20 ppt) were monitored.

### 2.8. Anesthetic Concentration and Recovery Time

Shrimp behaviors were used to evaluate the appropriate dose for anesthetic induction and recovery. In this study, 10 whiteleg shrimp were reared in 13 plastic tanks (10 L capacity) containing different concentrations of the STD-CO and CO-NLCs (10, 20, 30, 40, 50, 60, 70, 100, 120, 140, 160, 180, and 200 ppm) in triplicates. The classification of anesthesia and recovery stages are represented in Table 1 [21].

The induction and recovery times were recorded by one observer per stage. The treatment was stopped after the shrimp showed a complete loss of equilibrium (induction time stage 2). The shrimp were transferred to the recovery tank and observed until full recovery in stage 2.

### 2.9. Acute Toxicity Study of Clove Oil NLC in Shrimp

The acute toxicity test was conducted by following the method of Cansian, et al. [29] with slight modifications. After an acclimatization period, ten shrimp were stocked in a 10-L capacity tank. The toxicity test of the STD-CO and CO-NLCs was performed with shrimps, which included nineteen experimental concentrations (30, 40, 50, 60, 70, 100, 120, 140, 160, 180, 200, 250, 300, 350, 400, 450, 500, 550, and 600 ppm) and one control with triplicates. The mortality rate was determined at 72 h after being given the anesthesia.

### 2.10. Biodistribution Study of Clove Oil on Shrimps

In this experiment, we investigated the biodistribution of fluorescently labeled clove oil in shrimp bodies visualized by bioluminescence imaging [30]. Shrimps were divided into three groups: control (untreated), STD-CO, and CO-NLCs. Ten shrimp were immersed in 50 ppm of the STD-CO and CO-NLCs in triplicates. An individual shrimp was harvested from the tank when it reached the onset of stage 2. The shrimp were immediately transferred to a well-oxygenated recovery tank. After full recovery, the shrimps were collected every 2 min to evaluate excretion time as the shrimp eliminated clove oil particles. The fluorescent signal of the Nile Red-stained nanoparticles was examined using fluorescence imaging and the images were analyzed by IMAGE J software. Non-immersed shrimp were used as a background subtraction control. The fluorescent signals measured using in vivo imaging correlated with the amount of administered clove oil that accumulated in the shrimp body. Moreover, the whole shrimp fluorescence imaging was used to monitor the in vivo distribution of nanoparticles and the elimination of clove oil from the shrimp body.

### 2.11. Statistical Analysis

Data were expressed as the mean ± standard deviation (SD) for each group, and the statistical analyses were performed using SPSS (Version 26). The student *t*-test was used to compare the statistical difference between the means of the STD-CO and CO-NLCs. The Pearson’s correlation coefficients (r) were obtained for different concentrations or times of the STD-CO or CO-NLCs. LC_50_ was calculated from Probit analysis. Kaplan–Meier survivorship curves and a log-rank (Mantel–Cox) test were used to compare the groups based on the cumulative survival percentage. A value of *p* < 0.05 was considered statistically significant.

## 3. Results

### 3.1. Physicochemical Characterization of CO-NLCs

The clove oil was insoluble in water (Appendix A) and CO-NLCs were observed to be easily dissolved in water without phase separation (Appendix A), and appearances were in yellow transparent and milky translucent colors (Appendix A), respectively. The average particle size, PDI, and zeta potential value of the CO-NLCs were 175.07 ± 0.72 nm, 0.115 ± 0.230, and −48.37 ± 0.38 mV, respectively (Table 2).

The aggregation or accumulation of CO-NLCs were not shown in the images of the transmission electron microscope (TEM). The spherical CO-NLCs (Figure 2A) were assessed in the nanometer size range when harmonized with the dynamic light scattering measurement results (Figure 2B1,B2).

The CO-NLCs were stable for up to 3 months of storage in different conditions (30 °C/60% RH and 40 °C/75% RH) (Appendix A). The CO-NLCs particle size (Figure 3A) and zeta potential (Figure 3B) for 1, 2, and 3 months did not differ *(p* < 0.05) compared to the initial size and zeta potential in either condition. 

### 3.2. In Vitro Drug Release

The average encapsulation efficiency of CO-NLCs was 88.55%. The STD-CO and CO-NLCs showed a significantly (*p* < 0.01) very strong positive (R^2^ = 0.915 and 0.926) correlation of releasing ability with time (Figure 4).

The result indicates that the release profiles of the STD-CO exhibited a burst release. Eugenol was immediately released in a few minutes and reached the 30% level within 1 h, which was significantly higher than the CO-NLCs (*t*_2.113_ = 7.509, *p* < 0.05). In contrast, the CO-NLCs were able to sustain the release of eugenol. Only 5% of eugenol was gradually released from the CO-NLCs in the first hour. In addition, less than 20% of eugenol was released and detected after 2 h, which was significantly lower than the STD-CO (*t*_3.619_ = 3.999, *p* < 0.05). The linear regression trendline of drug release appeared between 2 to 8 h duration.

### 3.3. Effective Concentration for Induced Anesthesia

The STD-CO and CO-NLCs showed a significantly (*p* < 0.01) strong negative correlation of induction (R^2^ = −0.706 and −0.852) and a very strong positive correlation of recovery time (R^2^ = 0.946 and 0.912) with concentrations. Both the STD-CO and CO-NLCs were able to induce anesthesia in shrimp that entered stage 2 without complications. The 20 ppm was the lowest concentration of the STD-CO and CO-NLCs that allowed shrimp to reach stage 2 (Figure 5A). Interestingly, the STD-CO needed a significantly higher (31 min) induction time than the CO-NLCs (*t*_2.095_ = 111.534, *p* < 0.05) with 6.30 min. However, the induction time of the STD-CO treatment decreased to 12 (*t*_2.022_ = 27.498, *p* < 0.05) and 5 min (*t*_2.160_ = 24.122, *p* < 0.05) at 30 and 40 ppm concentration, respectively. Furthermore, for medium concentrations (30–70 ppm), the CO-NLCs had a significantly (*p* < 0.05) shorter duration of induction time than the STD-CO. The higher concentrations (100–200 ppm) of the STD-CO and CO-NLCs demonstrated rapid anesthetic effects on shrimp within 0.85–3 min after immersion in the CO-contained tanks.

The STD-CO and CO-NLCs treatments with 20 ppm concentrations did not differ in the recovery times (3.50 and 3 min) for shrimps from the complete loss of equilibrium stage (Figure 5B). Similarly, there was no significant difference (*p* > 0.05) in recovery time for medium concentrations (30–70 ppm) between the STD-CO and CO-NLCs. The higher concentration of the STD-CO resulted in a longer recovery duration than the CO-NLCs. With an increasing concentration of clove oil, the duration required to achieve sedation and anesthesia decreased, and the recovery times increased for the STD-CO shrimp in contrast to the CO-NLCs.

### 3.4. Toxicity of Clove Oil in Whiteleg Shrimp

The shrimp survival in the STD-CO and CO-NLCs reported a significantly (*p* < 0.01) strong (R^2^ = −0.791) and moderate (R^2^ = −0.436) negative correlation with concentrations, indicating the higher concentrations can increase the mortality rate. The median lethal concentrations (LC_50_) of the STD-CO and CO-NLC calculated from the Probit regression analysis were 143.8 ± 2 ppm and 353.1 ± 16.4 ppm, respectively. The log-rank (Mantel-Cox) test from the Kaplan–Meier analysis found statistically significant differences in the cumulative survival percentage for all the concentrations of the STD-CO (X^2^(19) = 473.373, *p* < 0.01) and CO-NLC (X^2^(19) = 343.881, *p* < 0.01) (Figure 6A,B). The toxicity results showed a concentration-dependent mortality in the STD-CO (>120 ppm) and CO-NLC (>140 ppm) groups. Within 24 h of post-anesthesia, the lowest and highest mortality rates in the STD-CO were observed at 140 ppm (50%) and 300 ppm (100%) (Figure 6A). Moreover, in the CO-NLC, the mortality of whiteleg shrimp was highest at 550 ppm (100%) and lowest at 160 ppm (10%) (Figure 6B).

### 3.5. Biodistribution Study of Clove Oil in Shrimps

The biodistribution of the STD-CO and CO-NLCs of the shrimp was examined through fluorescence signal imaging on the shrimp body at different times after full recovery. The fluorescence intensity was represented by a multicolor range from green (low intensity of CO) to red (high intensity of CO). From whole-body imaging (Figure 7), the CO-NLC was removed from the shrimp body within 30 min after recovery, while the STD-CO was still detected. 

The quantification of fluorescence intensity of whole-body imaging using ImageJ software confirmed the presence of CO accumulation. The results showed that the excretion of the CO-NLCs was faster than the STD-CO (Figure 8). In 30 min, the shrimp excreted the CO-NLCs but almost 20% (2000/12,000) of the STD-CO remained in the shrimp body.

## 4. Discussion

The use of anesthetics such as MS-222 and Aqui-SM in aquaculture has been approved for human consumption in several countries [21]. The concentrations needed to anesthetize crustaceans, however, seem to be relatively higher than those needed for fish [31,32]. Cooling and carbon dioxide are efficient anesthetic methods for crustaceans, but such approaches are expensive and inconvenient for live transport [33]. In recent years, essential oils such as eugenol and menthol have been used as cheap and safe anesthetics in many crustaceans [21]. Importantly, our group successfully developed and evaluated a safe, cost-effective, and easy-to-use form of clove essential oil by converting the poorly water-soluble CO into a soluble nanoparticulate platform for tilapia anesthesia [25].

In the present study, we investigated the physicochemical characterization of the CO-NLCs, a new lipid nanoparticle-carrier system of clove oil that is strong enough to be used as an alternative platform for anesthetic agents in shrimp. In general, STD-CO is insoluble in water. Therefore, STD-CO needs to be dissolved and diluted in ethanol before it can be used as an anesthetic. In drug-delivery systems, PDI values ≤ 0.3 (Table 2) are acceptable since it explicates a homogeneous population of phospholipid nanoparticles [34,35,36]. Our study showed that the CO-NLCs had a small droplet size (175 nm) with a low PDI (0.12), indicating that the system has a relatively narrow size distribution (Table 2). Similar findings have been reported in clove oil nanocarriers [37]. A zeta potential value of ±30 mV indicates that a nanodroplet or nanoparticle formulation is considered stable [38]. The stability of the CO-NLC formulation was consistent at 30 °C (60% RH) and 40 °C (75% RH) for up to 3 months, with a high negative zeta potential value (−48 mV) of particles (Figure 3A,B). It indicates the long-term stability of the dispersion with a lower chance of aggregation and refers to the degree of electrostatic repulsion between particles in a dispersion [39]. The observed CO-NLC particles were uniform and spherically shaped with no agglomeration or aggregation, which is consistent with clove oil nanocarriers [37]. Moreover, the particle size was measured to be in the nanometer range, which corresponds with the results from the DLS measurement. 

Regarding the release profile pattern, clove oil adsorbed on the surface was responsible for the initial burst of release when it became accessible at the interface, while the embedded clove oil was responsible for the persistent release [40]. The STD-CO showed an uncontrollable and irregular pattern of the cumulative release profile and a higher cumulative percentage release than the CO-NLCs (Figure 4). The negative charges of the CO-NLC particles absorbed into the emulsifier layer of oil, water interface, and electric double layer [41,42] could impact the anesthetic effect of clove oil. Moreover, the CO-NLCs have a high encapsulation efficiency (88.55%), possibly because of their lipophilic nature and higher loading capacity in stabilized formulations [37]. The result suggests that the new generation of lipid carrier systems is effective in facilitating the controlled release of clove oil and ensuring the superiority of lipid-based delivery nanostructured lipid carriers.

The biological characteristics of clove oil to be used as an anesthetic in immersion mode have also been assessed in this study. The STD-CO showed that the increased exposure period of clove oil resulted in a longer recovery time (Figure 5A). Similar findings have been reported for clove oil anesthetics in *Piaractus brachypomus* [43]. Importantly, the anesthetic efficacy is dependent on the eugenol’s solubility in lipids, as solubility enables the eugenol to pass through the cell wall of the gills [24,44]. Coating anatomic structures, especially gill epithelial cells, with oily clove oil or eugenol could lead to prolonged anesthetic effects. The induction time of STD-CO coincides with a study of the anesthetic effect of clove oil in *P. semisulcatus* [14]. The extremely small size, with a large interfacial area of internal oil droplets-nano carrier and water miscibility of insoluble ingredients [45,46], appears to be responsible for the rapid absorption of clove oil through the gills. This is in agreement with our study, which clearly indicated that a shorter induction time and rapid recovery were achieved with clove oil NLCs in shrimp in a dose–dependent manner (Figure 5B).

The mechanisms of CO-NLCs in shrimps are not well known. The anesthetic response could be regulated by γ-Aminobutyric acid type A (GABA_A_) neurotransmitters and metabotropic glutamate receptors, both of which have been identified in certain invertebrates [47,48,49]. In addition, anesthesia may inhibit neural and motor activity in the animals by blocking voltage-dependent K+ and Na+ channels [50]. Furthermore, the previous studies reported that eugenol and isoeugenol can activate the GABA_A_ receptor in the central nervous system [30,51,52]. Therefore, we infer that the small particle and lipophilic nature of CO-NLCs have a tranquilizing effect on the brain and thereby potentiate the GABA_A_ receptors’ response. Thus, additional study is needed to determine how CO-NLCs activate the GABA_A_ receptor response.

It was noted that when encapsulated in the NLC complex, eugenol compounds displayed significantly lower LC_50_ values (353 ppm) than those of the STD-CO (144 ppm) concerning the biodistribution and excretion results (Figure 6A,B). A similar result was reported in the toxicity of ginger oil NLC [53]. As a result, the nanoparticles have been proven to have a protective effect against the toxicity of the active compound in clove oil, yet their anesthesia effect has been maintained [54]. Whiteleg shrimp in the STD-CO group die rapidly in the concentration range of 120–200 ppm. This is probably because of the physical properties of clove oil, where it coats anatomic structures. Soltani et al. [14] reported that most prawns exposed to high concentrations of clove oil needed resuscitation and that the risk of ventilatory failure increased with increasing doses of clove oil. Moreover, this may prove to be important when it persists on gill epithelia, resulting in prolonged exposure to the chemical and the potential increase in sustained anesthetic effects, as mentioned by Sladky et al. [43]. Moreover, the survival rate of whiteleg shrimp in the CO-NLCs group basically remains above 80%. This is probably because the NLCs are excellent delivery systems, mainly due to their low toxicity [55]. In addition, Du et al. [56] reported that the increasing binding properties of neurotransmitter receptors with nanosized drug-delivery systems provide more effective and less toxic therapies.

The biodistribution of clove oil in the shrimp body was evaluated by the in vivo distribution of STD-CO and CO-NLCs in shrimp. The fluorescent signals measured using in vivo imaging correlated with the administered amount of clove oil that accumulated in the shrimp body. During the 30-min elimination kinetics experimental period, clove oil began to discrete at 4 min after shrimp recovery (Figure 8). The levels of eugenol declined rapidly in the shrimp treated with clove oil NLCs, but the elimination rate of eugenol was relatively slow in the STD-CO shrimp. The fluorescence image indicated that the clove oil level in the shrimp body was typically higher in the absorption and distribution phases. This study extended the examination to achieve the preferable accumulation of STD-CO in shrimp bodies by immersion administration since excessive accumulation of clove oil may cause toxic effects. The clove oil nanostructures degraded and were eliminated within 30 min, as seen in the mice [57]. Interestingly, the fluorescent signal was highly visible in the hepatopancreas and intestine of affected shrimp. This suggested that the intestinal tract was also the main biodistribution and excretory route for clove oil clearance in whiteleg shrimp apart from the gill. 

## 5. Conclusions

In the present study, we applied innovative nanotechnology to develop a drug-delivery system suitable to enhance the anesthetic activity of clove oil in shrimp by bath immersion. The CO-NLC formulation had a particle size of approximately 175 nm with a spherical shape, a PDI of 0.12, and a negative zeta potential of −48.37 mV that was stable for up to 3 months of storage. The average encapsulation efficiency of the CO-NLCs was 88.55%. In addition, the CO-NLCs were able to release 20% of eugenol after 2 h, which was lower than the STD-CO. The CO-NLC at 50 ppm observed the lowest sedation and anesthesia (2.2 min), the recovery time (3.3 min) and the rapid clearance (30 min) in shrimp body biodistribution. Our finding highlights a potential opportunity for NLC as a potent alternative nanodelivery platform which can be suitable for the enhanced anesthetic activity of clove oil in whiteleg shrimp. Further studies are needed to determine how the CO-NLCs activate the GABA_A_ receptor response and compare the anesthetic effect of the CO-NLCs with the chemical anesthetics for whiteleg shrimp. In addition, the long-term anesthetic effectiveness of the CO-NLCs for handling and transportation of farmed shrimp, as well as their physiological effects on shrimp, should be evaluated.

## Figures and Tables

**Figure 1 foods-11-03162-f001:**
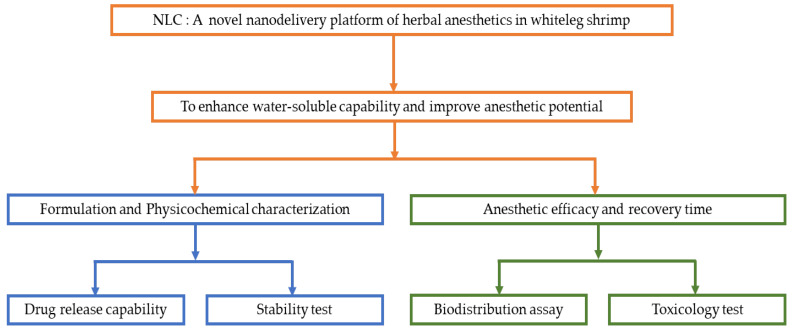
Schematic overview of the experimental program.

**Figure 2 foods-11-03162-f002:**
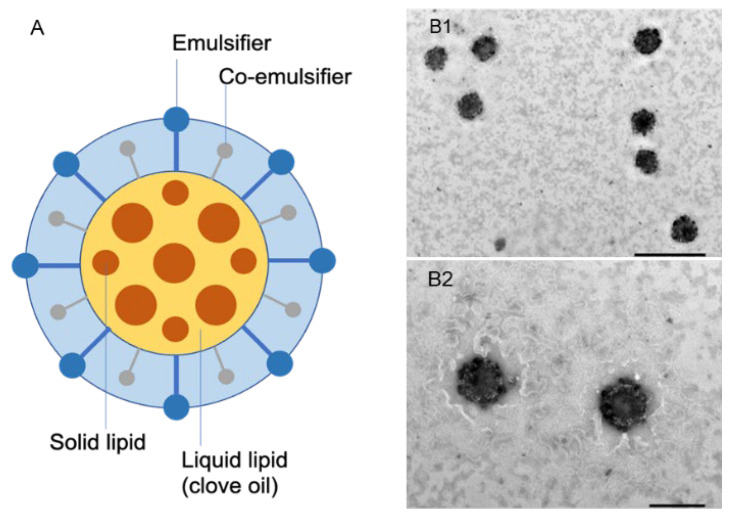
The schematic representation of CO-NLC (**A**). The morphology of CO-NLCs visualized by transmission electron microscopy (**B1**,**B2**). Scale bar: 200 nm (**B1**) and 100 nm (**B2**), respectively.

**Figure 3 foods-11-03162-f003:**
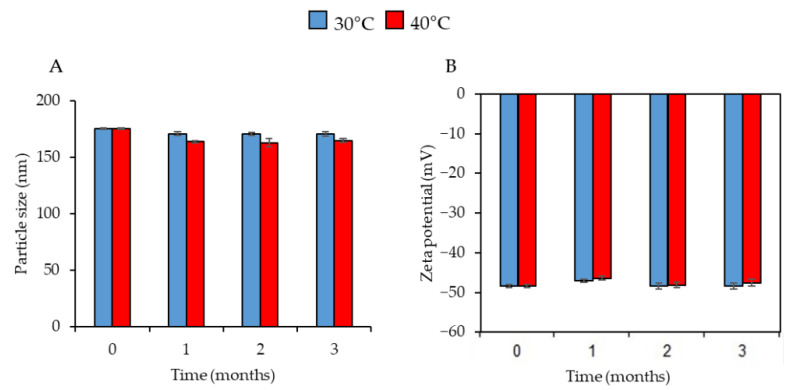
Particle size (**A**) and zeta potential (**B**) of CO-NLCs after storage at 30 °C/60% RH and 40 °C/75% RH for up to 3 months. No statistically significant difference of particle size and zeta potential (*p* < 0.05) were found when stored at 30 °C compared to 40 °C. Data are expressed as mean ± standard deviation (*n* = 3).

**Figure 4 foods-11-03162-f004:**
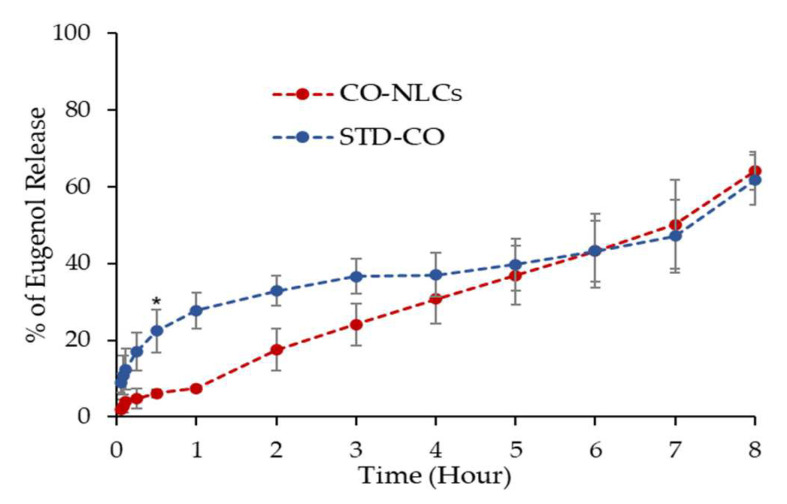
Eugenol release profile compared to the release profile from STD-CO and CO-NLCs at different timepoints. (*) indicates a significant difference (*t*-test, *p* < 0.05) between STD-CO and CO- NLCs for the particular time point.

**Figure 5 foods-11-03162-f005:**
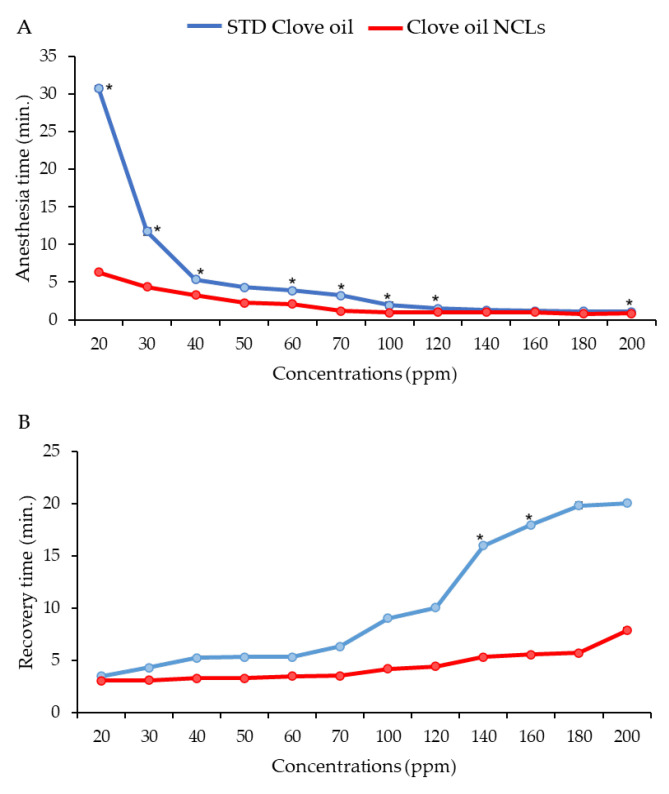
General anesthesia induction time in white shrimp after exposure to STD-CO and CO-NLCs (**A**). Recovery time from general anesthesia in the anesthetized white shrimp induced by STD-CO and CO-NLCs (**B**). (*) indicates a significant difference (*t*-test, *p* < 0.05) between STD-CO and CO-NLCs for the particular concentration.

**Figure 6 foods-11-03162-f006:**
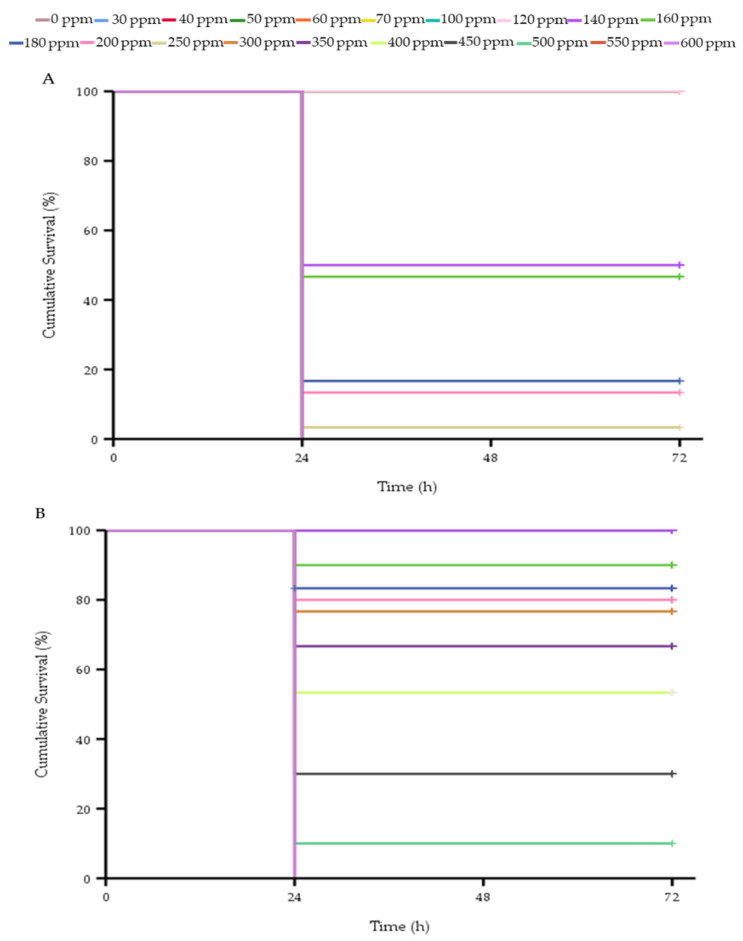
The cumulative survival percentage of whiteleg shrimp after anesthesia of STD-CO (**A**) and CO-NLC (**B**). Kaplan–Meier survivorship curves over time (h) for whiteleg shrimp was constructed and performed the log-rank (Mantel-Cox) statistical test to compare the groups.

**Figure 7 foods-11-03162-f007:**
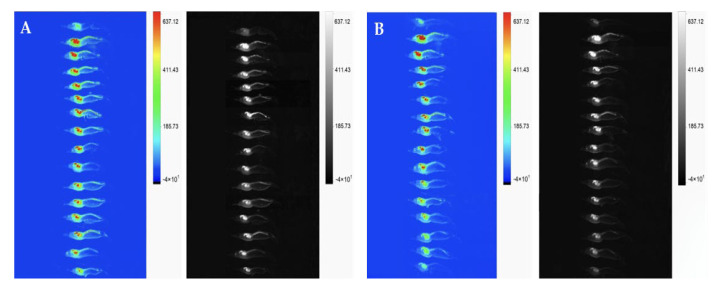
Clove oil biodistribution after exposure. RGB spectrum and gray scale of STD-CO (**A**) and CO-NLCs (**B**) in the whiteleg shrimp at 2 to 30 min after recovery. Images revealed the residual and excretion of CO using multicolor fluorescence distribution. High intensity CO appears as a red color, while low intensity CO appears as a green color.

**Figure 8 foods-11-03162-f008:**
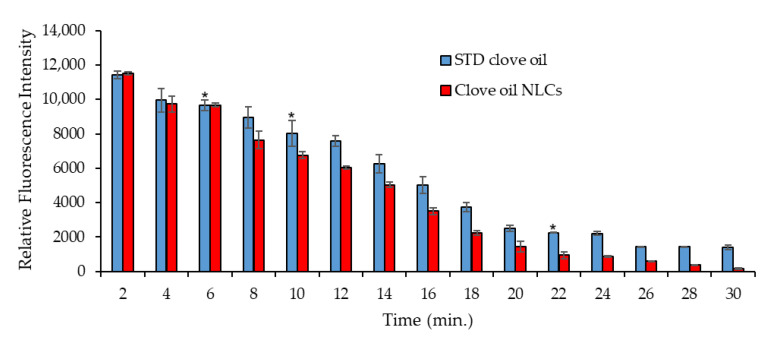
Quantification of CO accumulation and excretion in shrimp at different times after recovery. The comparison lines indicate a significant higher relative fluorescence intensity (*p* < 0.05) gained from the STD-CO compared to the CO-NLCs. (*) indicates a significant difference (*t*-test, *p* < 0.05) between STD-CO and CO-NLCs for the particular concentration.

**Table 1 foods-11-03162-t001:** Stages of anesthesia in shrimp according to Li et al. [21] (License number 5405151104023).

Stage	Behavior of Shrimp
Induction	
1	Reaction only to strong tactile and vibration stimuli
2	Non-reactivity to stimuli
Recovery	
1	Start of erratic swimming without reestablishment of equilibrium
2	Attained an upright position on the bottom of the aquaria

**Table 2 foods-11-03162-t002:** Physicochemical characterization of clove oil NLCs.

Formulation	Particle Size (nm)	Zeta Potential (mV)	Polydispersity Index	Appearance
STD Clove oil	NA	NA	NA	Yellow transparent
Clove NLCs	175 ± 0.72	−48.37 ± 0.38	0.115 ± 0.230	Milky translucent

NA: not applicable; STD: standard; NLC: nanostructured lipid carrier; nm: nanometer; mV: millivolts.

## Data Availability

The data presented in this study are available on request from the corresponding author.

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
