# Peer review of "Clove Oil-Nanostructured Lipid Carriers: A Platform of Herbal Anesthetics in Whiteleg Shrimp (Penaeus vannamei)"

_foods, 2022, doi:10.3390/foods11203162_

Round 1

Reviewer 1 Report

1.      Please analyze the mechanism of CO-NLCS applied to whiteleg shrimp anesthesia.

2.      Please expound the reasons for the slow release of CO-NLC compared with STD-CO in figure 3.

3.      In the manuscript, the release rate in CO-NLCs in Figure 3 is slower compared to STD-CO. What is the reason for the significant and lower anesthesia time of CO-NLCs than STD-CO for whiteleg shrimp in Figure 4A?

4.      In Figure 5, concentrations of STD-CO and CO-NLCs above 100 ppm cause whiteleg shrimp mortality. Whiteleg shrimp in the STD-CO group die rapidly in the concentration range of 120-200 ppm, but the survival rate of whiteleg shrimp in the CO-NLCs group basically remains above 80%, please analyze the reasons.

5.      Does the amount of clove oil encapsulated have an effect on the release of CO-NLCs? If so, please explain with experimental data.

6.      To promote the use of CO-NLCs in aquaculture, please compare the anesthetic effect of CO-NLCs with that of anesthetics (benzocaine, quinaldine, 2-phenoxyethanol (2-PE), and tricaine methanesulfonate (MS-222)) for whiteleg shrimp using experimental data.

7.      Several minor points should be corrected.

i.                    All abbreviations need to be defined for the first time they appear.

ii.                  The legend in Figure 6 is unclear.

Author Response

Response to Reviewer 1 Comments

Point 1: Please analyze the mechanism of CO-NLCS applied to whiteleg shrimp anesthesia.

Response: Thank you so much for this valuable comment. The mechanisms of CO-NLCs in shrimps are not well known. The anesthetic response could be regulated by GABA neurotransmitters and metabotropic glutamate receptors, both of which have been identified in certain invertebrates (Elwood, 2019; Hamilton et al., 2016; Perrot-Minnot et al., 2017). In addition, anesthesia may inhibit neural and motor activity in the animals by blocking voltage-dependent K+ and Na+ channels (Wycoff et al., 2018). Furthermore, the previous studies reported that eugenol and isoeugenol can activate the GABAA receptor in the central nervous system (Aoshima et al., 1999, Zahl et al., 2012, and Kheawfu et al., 2022). Therefore, we infer that the small particle and lipophilic nature of CO-NLCs have a tranquilizing effect on the brain and thereby potentiate the GABAA receptors' response. Thus, additional study is needed to determine how CO-NLCs activate the GABAA receptor response.

The above mechanism has been updated in the manuscript from lines 353-362.

References

Aoshima, H.; Hamamoto, K. Potentiation of GABAA receptors expressed in Xenopus Oocytes by perfume and phytoncid. Biosci. Biotechnol. Biochem. 1999, 63, 743–748.

Elwood, R. W. (2019). Discrimination between nociceptive reflexes and more complex responses consistent with pain in crustaceans. Philosophical Transactions of the Royal Society B374(1785), 20190368.

Hamilton, T. J., Kwan, G. T., Gallup, J., & Tresguerres, M. (2016). Acute fluoxetine exposure alters crab anxiety-like behaviour, but not aggressiveness. Scientific reports6(1), 1-6.

Perrot-Minnot, M. J., Banchetry, L., & Cézilly, F. (2017). Anxiety-like behaviour increases safety from fish predation in an amphipod crustacea. Royal Society Open Science4(12), 171558.

Wycoff, S., Weineck, K., Conlin, S., Suryadevara, C., Grau, E., Bradley, A., ... & Cooper, R. L. (2018). Effects of clove oil (eugenol) on proprioceptive neurons, heart rate, and behavior in model crustaceans. Impulse2018, 1.

Zahl, I.H.; Samuelsen, O.; Kiessling, A. Anaesthesia of farmed fish: Implications for welfare. Fish Physiol. Biochem. 2012, 38, 201–218.

Point 2: Please expound the reasons for the slow release of CO-NLC compared with STD-CO in figure 3.

Response: Thank you for the valuable comment. The release of the encapsulated compound loaded into NLCs is based on the active compound diffusing from the NLCs into the surrounding environment before diffusing through the dialysis membrane. In contrast, STD-CO, which was dissolved in ethyl alcohol, diffused directly through the dialysis membrane. Therefore, STD-CO provided a burst release of an active compound compared to CO-NLCs.

Point 3: In the manuscript, the release rate in CO-NLCs in Figure 3 is slower compared to STD-CO. What is the reason for the significant and lower anesthesia time of CO-NLCs than STD-CO for whiteleg shrimp in Figure 4A?

Response: Thank you for the excellent observation. The release of the encapsulated compound loaded into NLCs is based on the active compound diffusing from the NLCs into the surrounding environment before diffusing through the dialysis membrane. In contrast, STD-CO, which was dissolved in ethyl alcohol, diffused directly through the dialysis membrane. Therefore, STD-CO provided a burst release of an active compound compared to CO-NLCs. However, the CO-NLCs had a significantly shorter anesthetic duration due to their higher systemic and cellular uptake compared to STD-CO.

References

Chaudhari, V. S., Gawali, B., Saha, P., Naidu, V. G. M., Murty, U. S., & Banerjee, S. (2021). Quercetin and piperine enriched nanostructured lipid carriers (NLCs) to improve apoptosis in oral squamous cellular carcinoma (FaDu cells) with improved biodistribution profile. European Journal of Pharmacology909, 174400.

Varela-Fernández, R., García-Otero, X., Díaz-Tomé, V., Regueiro, U., López-López, M., González-Barcia, M., ... & Otero-Espinar, F. J. (2022). Lactoferrin-loaded nanostructured lipid carriers (NLCs) as a new formulation for optimized ocular drug delivery. European Journal of Pharmaceutics and Biopharmaceutics172, 144-156.

Point 4: In Figure 5, concentrations of STD-CO and CO-NLCs above 100 ppm cause whiteleg shrimp mortality. Whiteleg shrimp in the STD-CO group die rapidly in the concentration range of 120-200 ppm, but the survival rate of whiteleg shrimp in the CO-NLCs group basically remains above 80%, please analyze the reasons.

Response: Thank you so much for the excellent observation.

Whiteleg shrimp in the STD-CO group die rapidly in the concentration range of 120-200 ppm. This is probably because of the physical properties of clove oil, where it coats anatomic structures. Soltani et al. (2004) reported that most prawns exposed to high concentrations of clove oil needed resuscitation and that the risk of ventilatory failure increased with increasing doses of clove oil. Moreover, this may prove to be important when it persists on gill epithelia, resulting in prolonged exposure to the chemical and the potential increase of sustained anesthetic effects, as mentioned by Sladky et al. (2001).

But the survival rate of whiteleg shrimp in the CO-NLCs group basically remains above 80%. This is probably because the NLC are excellent delivery systems, mainly due to their low toxicity (Akhavan et al., 2018). In addition, Du et al. (2014) reported that the increasing binding properties of neurotransmitter receptors with nanosized drug-delivery systems provides more effective and less toxic therapies.

The above reasons have been updated in the manuscript from lines 368-379.

References

Akhavan, S., Assadpour, E., Katouzian, I., & Jafari, S. M. (2018). Lipid nano scale cargos for the protection and delivery of food bioactive ingredients and nutraceuticals. Trends in Food Science & Technology74, 132-146.

Du, H., Yang, X., & Zhai, G. (2014). Design of chitosan-based nanoformulations for efficient intracellular release of active compounds. Nanomedicine9(5), 723-740.

Sladky, K. K., Swanson, C. R., Stoskopf, M. K., Loomis, M. R., & Lewbart, G. A. (2001). Comparative efficacy of tricaine methanesulfonate and clove oil for use as anesthetics in red pacu (Piaractus brachypomus). American journal of veterinary research62(3), 337-342.

Soltani, M., Marmari, G. H., & Mehrabi, M. R. (2004). Acute toxicity and anesthetic effects of clove oil in Penaeus semisulcatus under various water quality conditions. Aquaculture International12(4), 457-466.

Point 5: Does the amount of clove oil encapsulated have an effect on the release of CO-NLCs? If so, please explain with experimental data.

Response: Thank you for your comment. Prior to determining encapsulation efficiency, clove oil loading efficiency in NLCs was optimized to maximize particle stability. Moreover, encapsulation efficiency plays an important role, which can affect the release profile of active compounds from NLCs. At high encapsulated concentrations, the release rate tends to increase due to the concentration gradient between the release buffer and encapsulation particle. However, particle size, lipid composition, and surfactant to oil ratio (SOR) are the other factors that affect the release mechanism of NLCs.

Reference

Teeranachaideekul, V., Boonme, P., Souto, E. B., Müller, R. H., & Junyaprasert, V. B. (2008). Influence of oil content on physicochemical properties and skin distribution of Nile red-loaded NLC. Journal of controlled release128(2), 134-141.

Point 6: To promote the use of CO-NLCs in aquaculture, please compare the anesthetic effect of CO-NLCs with that of anesthetics (benzocaine, quinaldine, 2-phenoxyethanol (2-PE), and tricaine methanesulfonate (MS-222)) for whiteleg shrimp using experimental data.

Response: Thank you for your comment and suggestion. As the reviewer suggested, we have a time limit to conduct a new experiment to compare the anesthetic effect of CO-NLCs with that of chemical anesthetics. Thus, we would like to recommend this for further study.

The above recommendation has been updated in the manuscript from lines 406-410.

Point 7: Several minor points should be corrected.

Point 7.1: All abbreviations need to be defined for the first time they appear.

Response: Thank you for your comment. We have made necessary changes in the manuscript.

Point 7.2: The legend in Figure 6 is unclear.

Response: Thank you for your comment. We have revised the legend of Figure 6 and remaining text of the legend has been added in the Material and Method section in the manuscript.

Lines 287-290: Figure 7. Clove oil biodistribution after exposure. RGB spectrum and gray scale of STD-CO (A) and CO-NLCs (B) in the whiteleg shrimp at 2 to 30 min after recovery. Images revealed the residual and excretion of CO using multicolor fluorescence distribution. High intensity CO appears as a red color, while low intensity CO appears as a green color.

Lines 180-184: Non-immersed shrimp were used as a background subtraction control. The fluorescent signals measured using in vivo imaging correlate with the amount of administered clove oil which accumulated in the shrimp body. Moreover, the whole shrimp fluorescence imaging was used to monitor the in vivo distribution of nanoparticles and the elimination of clove oil from the shrimp body.

Reviewer 2 Report

The manuscript entitled "Clove Oil-Nanostructured Lipid Carriers: A Platform of Herbal 2 Anesthetics in Whiteleg Shrimp (Penaeus vannamei)" shows the utilization of nanostructured lipid carriers carrying Clove oil in anesthesia procedures. This manuscript is valuable. However, the authors should consider the following things:

1/ What is "STD-CO"? and how did the authors prepare it?

2/ how long does the effect from CO-NLC last?

3/ What is the negative control in LC50 assay? the authors should conduct Probit analysis for LC50

4/ The authors should do the Kaplan–Meier analysis and provide Kaplan-Meier curve of survival rate at concentration of 200ppm

Author Response

Response to Reviewer 2 Comments

The manuscript entitled "Clove Oil-Nanostructured Lipid Carriers: A Platform of Herbal 2 Anesthetics in Whiteleg Shrimp (Penaeus vannamei)" shows the utilization of nanostructured lipid carriers carrying Clove oil in anesthesia procedures. This manuscript is valuable. However, the authors should consider the following things:

Point 1: What is "STD-CO"? and how did the authors prepare it?

Response: Thank you for your comment. The STD-CO means standard clove oil and it was prepared by mixing clove oil and absolute ethanol with respective volumetric ratio of 2:8.

Point 2: How long does the effect from CO-NLC last?

Response: Thank you so much for your comment. We had conducted a short-term study to determine the anesthetic time for whiteleg shrimp. The anesthetic drugs can stay in the system even after recovery time without any complications. The recommended dose of 50 ppm CO-NLCs required 2.2 min and 3.3 min of anesthesia and recovery time, respectively. Furthermore, from whole-body imaging, CO-NLC was removed from the shrimp body within 30 minutes after recovery. As we understood the reviewer's comments correctly, even after the recovery time of 3.3 min, the CO-NLC effect may last for another 25–30 min until complete excretion from the shrimp body.

Point 3: What is the negative control in LC50 assay? the authors should conduct Probit analysis for LC50.

Response: Thank you so much for your comment. Normal tank water was the negative control in the LC50 analysis. As per the suggestion of the reviewer to conduct Probit analysis for LC50, we have conducted a new experiment to determine the toxicity of STD-CO and CO-NLC with shrimps, which included nineteen experimental concentrations (30, 40, 50, 60, 70, 100, 120, 140, 160, 180, 200, 250, 300, 350, 400, 450, 500, 550, and 600 ppm) and one control with triplicates. The LC50 of STD-CO and CO-NLC from the Probit analysis was found to be 143.8±2 ppm and 353.1±16.4 ppm, respectively.

The above LC50 values have been updated in the manuscript from lines 264-266.

Point 4: The authors should do the Kaplan–Meier analysis and provide Kaplan-Meier curve of survival rate at concentration of 200ppm

Response: Thank you so much for your comment. We have conducted a new experiment to determine the toxicity of STD-CO and CO-NLC with shrimps, which included nineteen experimental concentrations (30, 40, 50, 60, 70, 100, 120, 140, 160, 180, 200, 250, 300, 350, 400, 450, 500, 550, and 600 ppm) and one control with triplicates. The log-rank (Mantel-Cox) test from the Kaplan-Meier analysis found statistically significant differences in the cumulative survival percentage for all the concentrations of STD-CO (X2(19) = 473.373, p < 0.01) and CO-NLC (X2(19) = 343.881, p < 0.01) (Figure 6A and B). The toxicity results showed a concentration-dependent mortality in the STD-CO (> 120 ppm) and CO-NLC (> 140 ppm) groups. Within 24 hours of post-anesthesia, the lowest and highest mortality rates in STD-CO were observed at 140 ppm (50%) and 300 ppm (100%). Also, in NLC-CO, the mortality of whiteleg shrimp was highest at 550 ppm (100%) and lowest at 160 ppm (10%) (Figure 6B).

Figure 6: The cumulative survival percentage of whiteleg shrimp after anesthesia of STD-CO (A) and CO-NLC (B). Kaplan-Meier survivorship curves over time (h) for whiteleg shrimp was constructed and performed the log-rank (Mantel-Cox) statistical test to compare the groups.

The above Kaplan-Meier survivorship results have been updated in the manuscript from lines 266-278.

Reviewer 3 Report

This is a very promising study, which investigated clove oil-nanostructured lipid carriers, specifically targeting it as a platform of herbal anesthetics in Whiteleg shrimp.

Overall, the manuscript reads well, and can be further strengthened by the following:

a) Introduction, please before paragraph on cloves, kindly provide a new paragraph discussing relevant herbs documented with anesthetic potentials. List key ones of Asia, make sure to mention cloves, discuss their availability and production, and geo-distribution. So when the next paragraph starts with clove, it will make sense to the reader.

Please, try to, in the last paragraph of introduction, provide information as to why this current work is relevant. Identify with near related studies, it might not necessarily be on shrimp, at least on a seafood , it might not be cloves, but at least a herbal plant, mentioned in the paragraph above, that has attempted a study , which helps to lay a foundation to this work. Build a strong argument ok

b) Materials and methods is ok, please reviewer suggests creating a new sub section, captioned ‘schematic overview of the experimental program ‘, that comprise at least 4 sentences, and supports compulsorily with flow diagram. Show succinctly how this work was planned, mentioned key aspects of the experimental activities, and put it in the context of the specific objective of this study. Make sure to reiterate the explanation of the objective of this work through this sub-section .

c) Both results and discussion are very promising. Authors are encouraged to provide more elaboration on the results. Authors are also encouraged to , in the discussion, identify with all the Tables and Figures that have been mentioned in the results section, all must be captured in the discussion . So, reviewer would look out for (Refer to Table?) and (Refer to Figure?) in the discussion section. 

d) Conclusion is ok, the reviewer suggests authors brainstorm and provide direction for future studies. What are also the limitations encountered in this work. Please mention

Looking forward to the revised manuscript. Thanks 

Author Response

Response to Reviewer 3 Comments

This is a very promising study, which investigated clove oil-nanostructured lipid carriers, specifically targeting it as a platform of herbal anesthetics in Whiteleg shrimp.

Overall, the manuscript reads well, and can be further strengthened by the following:

Point 1: Introduction, please before paragraph on cloves, kindly provide a new paragraph discussing relevant herbs documented with anesthetic potentials. List key ones of Asia, make sure to mention cloves, discuss their availability and production, and geo-distribution. So when the next paragraph starts with clove, it will make sense to the reader.

Please, try to, in the last paragraph of introduction, provide information as to why this current work is relevant. Identify with near related studies, it might not necessarily be on shrimp, at least on a seafood, it might not be cloves, but at least a herbal plant, mentioned in the paragraph above, that has attempted a study, which helps to lay a foundation to this work. Build a strong argument ok

Response: Thank you so much for your excellent suggestion.

Herbal anesthetics such as basil, thyme, mint, rosemary, lavender, citronella, verbena, and camphor have been used in aquaculture, although clove oil is the most often employed (Aydın et al., 2020). In Asia, essential oils from Mentha piperita (Mazandarani and Hoseini, 2018), Mentha spicata (Roohi and Imanpoor, 2015), Eucalyptus sp., and Origanum sp. (Bodur et al., 2018), and Syzygium aromaticum (Soltani et al., 2004, Hoseini and Nodeh, 2013) have been found to induce anesthesia in fish and shrimp with positive health effects. M piperita (Europe and the Middle East), M spicata (Europe and Asia), Origanum sp. (south-west Asia), S. aromaticum (south-east Asia), and Eucalyptus sp. (Australia) are available in their native areas and have been adapted to many regions of the world (Hoseini et al., 2019). The above description has been updated in the manuscript from lines 54-62.

Chemical anesthetics have limited application owing to safety issues in humans and fish (Gholipourkanani et al., 2015). In contrast to certain anesthetics such as MS-222, clove oil anesthesia does not need a withdrawal phase since it is rapidly excreted from blood and tissues (Ross and Ross, 2009). Consequently, clove oil seems to be a better alternative in commercial aquaculture operations, where anesthetics may be employed in substantial quantities by unskilled workers and discharged into natural water bodies. Clove oil, on the other hand, is insoluble in water. Prior to use for emulsification, it should be mixed with ethanol, which could be harmful to fish and other aquatic animals. The above description has been updated in the manuscript from lines 70-77.

References

Aydın, B., & Barbas, L. A. L. (2020). Sedative and anesthetic properties of essential oils and their active compounds in fish: A review. Aquaculture520, 734999.

Bodur, T., Afonso, J. M., Montero, D., & Navarro, A. (2018). Assessment of effective dose of new herbal anesthetics in two marine aquaculture species: Dicentrarchus labrax and Argyrosomus regius. Aquaculture482, 78-82.

Gholipourkanani, H., Gholinasab-Omran, I., Ebrahimi, P., & Jafaryan, H. (2015). Anesthetic effect of clove oil loaded on lecithin based nano emulsions in gold fish, Carassius auratus. Journal of Fisheries and Aquatic Science10(6), 553.

Hoseini, S. M., & Nodeh, A. J. (2013). Changes in blood biochemistry of common carp Cyprinus carpio (Linnaeus), following exposure to different concentrations of clove solution. Comparative Clinical Pathology22(1), 9-13.

Hoseini, S. M., Taheri Mirghaed, A., & Yousefi, M. (2019). Application of herbal anaesthetics in aquaculture. Reviews in Aquaculture11(3), 550-564.

Mazandarani, M., & Hoseini, S. M. (2017). Anesthesia of juvenile Persian sturgeon, Acipenser persicus; Borodin 1897, by peppermint, Mentha piperita, extract–Anesthetic efficacy, stress response and behavior. International Journal of Aquatic Biology5(6), 393-400.

Roohi, Z., & Imanpoor, M. R. (2015). The efficacy of the oils of spearmint and methyl salicylate as new anesthetics and their effect on glucose levels in common carp (Cyprinus carpio L., 1758) juveniles. Aquaculture437, 327-332.

Ross, L.G. and B. Ross, 2009. Anaesthetic and Sedative Techniques for Aquatic Animals. 3rd Edn., Wiley-Blackwell, New York, USA., ISBN-13: 9781405149389, Pages: 240.

Point 2: Materials and methods is ok, please reviewer suggests creating a new sub section, captioned ‘schematic overview of the experimental program ‘, that comprise at least 4 sentences, and supports compulsorily with flow diagram. Show succinctly how this work was planned, mentioned key aspects of the experimental activities, and put it in the context of the specific objective of this study. Make sure to reiterate the explanation of the objective of this work through this sub-section.

Response: Thank you so much for your excellent suggestion.

Figure 1: Schematic overview of the experimental program

Schematic overview of the experimental program

Prior to use as an anesthetic, clove oil should be mixed with ethanol, which may be detrimental to aquatic animals after discharging into natural water bodies. Also, it could be a skin allergy to unskilled labor. Therefore, we proposed a safe, cost-effective, and easy-to-use form of clove essential oil by converting the poorly water-soluble CO into a soluble novel nanodelivery platform of herbal anesthetics in whiteleg shrimp (Figure 1). To achieve this, we have formulated clove oil nanostructured lipid carriers (CO-NLC), followed by characterization, in vitro drug release capability, and stability tests. Importantly, the appropriate dose for anesthetic effect and biodistribution were investigated in the shrimp body as well as the acute multiple-dose toxicity study. CO-NLC could be a good alternative nano-delivery platform for increasing the anesthetic activity of clove oil in whiteleg shrimp. The above description has been updated in the manuscript from lines 101-113.

Point 3: Both results and discussion are very promising. Authors are encouraged to provide more elaboration on the results. Authors are also encouraged to, in the discussion, identify with all the Tables and Figures that have been mentioned in the results section, all must be captured in the discussion. So, reviewer would look out for (Refer to Table?) and (Refer to Figure?) in the discussion section. 

Response: Thank you so much for your comment and suggestions. By considering reviewers suggestion, we have provided the elaboration on the results wherever its necessary. Also, in the discussion we have included all the Tables and Figures that have been mentioned in the results section.

Point 4: Conclusion is ok, the reviewer suggests authors brainstorm and provide direction for future studies. What are also the limitations encountered in this work. Please mention.

Response: Thank you so much for your comment and suggestions.  

Further studies are needed to determine how CO-NLCs activate the GABAA receptor response and compare the anesthetic effect of CO-NLCs with chemical anesthetics for whiteleg shrimp. In addition, the long-term anesthetic effectiveness of CO-NLCs for handling and transportation of farmed shrimp, as well as their physiological effects on shrimp, should be evaluated. The above recommendations have been updated in the manuscript from lines 406-410.

The following limitations are encountered in this work:

  1. We had experienced skin irritation from non-encapsulated clove oil.
  2. The amount of active ingredient (eugenol) in clove oil was hard to control because it varies based on season, supplier, and cultivation area.
  3. We did not check the sedative dose effect, which is important for transportation.

 Point 5: Looking forward to the revised manuscript. Thanks 

Response: Thank you so much for your encouragement.  

Round 2

Reviewer 1 Report

Acceptable

Reviewer 3 Report

Thank you authors for the very impressive revisions carefully carried out.

The reviewer is very satisfied with the quality of the current version.

It is acceptable for publication.